# ROBUST FEDERATED LEARNING CLIENT SELECTION WITH COMBINATORIAL CLASS REPRESENTATIONS AND DATA AUGMENTATION

## ABSTRACT

The federated learning (FL) client selection scheme can effectively mitigate global model performance degradation caused by the random aggregation of clients with heterogeneous data. Simultaneously, research has exposed FL's susceptibility to backdoor attacks. However herein lies the dilemma, traditional client selection methods and backdoor defenses stand at odds, so their integration is an elusive goal. To resolve this, we introduce Grace, a resilient client selection framework blending combinational class sampling with data augmentation. On the client side, Grace first proposes a local model purification method, fortifying the model's defenses by bolstering its innate robustness. After, local class representations are extracted for server-side client selection. This approach not only shields benign models from backdoor tampering but also allows the server to glean insights into local class representations without infringing upon the client's privacy. On the server side, Grace introduces a novel representation combination sampling method. Clients are selected based on the interplay of their class representations, a strategy that simultaneously weeds out malicious actors and draws in clients whose data holds unique value. Our extensive experiments highlight Grace's capabilities. The results are compelling: Grace enhances defense performance by over 50% compared to state-of-the-art (SOTA) backdoor defenses, and, in the best case, improves accuracy by 3.19% compared to SOTA client selection schemes. Consequently, Grace achieves substantial advancements in both security and accuracy.

## 1 INTRODUCTION

Federated learning (FL) (McMahan et al. (2017)) has emerged as a novel distributed learning framework in which clients collaboratively train a high-performance machine learning model, coordinated by a centralized server, without sharing or exchanging private data. However, randomly aggregating local models with heterogeneous data can significantly degrade the global model's performance. Existing studies (Goetz et al. (2019); Cho et al. (2020); Nishio & Yonetani (2019); Yang et al. (2021b)) have found that selecting local models for aggregation based on the characteristics of clients' local models or data, i.e., using a heuristic client selection method instead of random selection, can effectively address the challenges posed by data heterogeneity.

Existing client selection studies can be divided into two categories: sample feature-based (Yang et al. (2021b); Fraboni et al. (2021); Balakrishnan et al. (2022); Zhang et al. (2023c)) and model feature-based methods (Cho et al. (2020); Han et al. (2023); Tang et al. (2022); Nagalapatti & Narayanam (2021); Yan et al. (2023)). Sample feature-based methods require clients to provide the server with information about the number of samples in their local training set or the distribution of sample categories. Clients with larger training sets or more balanced category distributions are then selected for aggregation. However, it is challenging for the server to verify the truthfulness of the number or distribution of samples provided by the clients. Consequently, current research has shifted focus to model feature-based approaches. This method selects representative and diverse local models for aggregation based on their losses, gradients, or parameters, aiming to enhance the generalization of the global model.

In addition, existing FL research has highlighted the vulnerability of Federated Learning (FL) to backdoor attacks (Hongyi et al. (2020); Bhagoji et al. (2019); Eugene et al. (2020); El Mahdi et al. (2018); Zhang et al. (2023a)), prompting the development of various defenses against such attacks. These defenses can be categorized into two main types: client-side (Zhu et al. (2023); Sun et al. (2021)) and server-side (Huang et al. (2023a); Han et al. (2023); Lycklama et al. (2023); Xie et al. (2021)). Client-side defenses aim to detect backdoor-related parameters in the local model that the server sends to the client, and then perturb or clip these parameters to defend against backdoor attacks. However, due to the diversity of client data and the large parameter space of the model, accurately identifying backdoor-related parameters remains a challenging task. As a result, server-side defenses have become the primary focus. These defenses work by comparing the similarity between the parameters of local models, identifying those that resemble most models as benign, and excluding dissimilar local models from the aggregation process to mitigate backdoor attacks.

However, we observe a conflict between existing client selection strategies and backdoor attack defenses. Specifically, the goal of client selection is to choose diverse local models, i.e., to select clients with unique data to participate in aggregation, thereby enhancing the generalization of the global model. In contrast, backdoor attack defense aims to exclude local models that differ from the majority, focusing on aggregating similar local models to mitigate backdoor threats. Local models trained on backdoor samples are challenging to identify and exclude within a client selection scheme based on model features. Additionally, clients with unique data characteristics may be mistakenly classified as malicious in server-side backdoor defenses. Consequently, existing backdoor defense mechanisms are challenging to integrate with client selection schemes. If the backdoor defense is not applied during the client selection process, local models embedded with backdoors will participate in global model aggregation, leading to vulnerabilities in the global model.

In response, we introduce Grace, a robust federated learning client selection with combinatorial class representations and data augmentation. Grace selects clients for aggregation using class representations instead of model features, which allows for excluding malicious clients while also selecting clients with unique data features. Given the challenge of excluding malicious models with 100% accuracy, we enhance the robustness of the local model internally on the client side. This method not only defends against attacks but also improves the accuracy of the global model. Grace is the first solution that defends against backdoor attacks without sacrificing model accuracy; instead, it enhances it, marking a significant breakthrough. Our contribution can be summarized as follows:

- We propose a local model purification method. First, we apply data augmentation techniques to construct augmented datasets, which are used to fine-tune the local initial models, enhancing their robustness. Next, we employ representation learning and mutual information to learn local class representations and align them with global class representations. This improves local model performance while also supporting server-side client selection. This approach not only ensures that the local models of benign clients remain free from backdoors but also allows the server to learn class representations of local samples without compromising client privacy.

- We propose a representation combinatorial sampling method. We first establish a metric to quantify the global class representation loss for each client and apply a combinatorial confidence upper bound algorithm to sample clients, excluding those with significant global class representation losses. After, an expert product technique is used to generate global class representations, which guide the learning of local class representations. This method effectively excludes malicious clients while selecting high-quality clients.

- We conduct a series of experiments to evaluate the effectiveness of Grace. Our experimental results demonstrate that Grace outperforms existing client selection methods across various heterogeneous settings. Moreover, Grace surpasses existing defenses under different attack scenarios and non-IID settings. Finally, ablation studies and hyperparameter analyses further validate the effectiveness of Grace's individual components.

## 2 RELATED WORK

**Client selection.** In prior literature, various endeavors have been undertaken to enhance client selection in FL. Methods such as Clustered (Fraboni et al. (2021)) rely on the number of client

samples for selection. However, this metric may not be appropriate, as clients could utilize a substantial amount of repetitive data, leading to potential overfitting of local models. Other approaches, including those by (Cho et al. (2020); Goetz et al. (2019); Tang et al. (2022)), select clients based on higher local losses. Nevertheless, this criterion doesn't guarantee a reduction in the final global model's losses. Nagalapatti & Narayanam (2021) introduces the assumption of a validation set for server-side validation of local model performance. However, obtaining a dataset that accurately models the global data distribution is often unrealistic. Researchers later identified a correlation between client diversity and gradients/parameters. Balakrishnan et al. (2022) selected clients by maximizing a submodule facility position function defined in the gradient space. CriticalFL (Yan et al. (2023)) analyzed parameter importance at different stages of FL for client selection. In recent work, Fed-CBS (Zhang et al. (2023c)) argues that the client selection process should not only consider diversity but also account for category imbalance.

**Backdoor attack.** In the context of FL, backdoor attacks (Hongyi et al. (2020); Bhagoji et al. (2019); Eugene et al. (2020); Chulin et al. (2020)) aim to manipulate global model predictions by exploiting compromised clients embedding backdoors. For instance, Eugene et al. (2020) introduced a scaling attack wherein the attacker utilizes a combination of backdoors and clean training samples to train its local model. Subsequently, the attacker scales the local model update before transmitting it to the server. In our research, we introduce a strategic backdoor attack, leveraging state-of-the-art attack techniques (Eugene et al. (2020); Zhang et al. (2023a); Chulin et al. (2020)), to assess the resilience of our defense mechanism.

**Backdoor defense.** In response to the increasing threat of poisoning attacks, current defense strategies involve restricting, removing updates, or adding noise to the norm. Notable defenses include SparseFed (Panda et al. (2022)), which addresses poisoning attacks by selectively updating the most relevant weights in an aggregation model. DeepSight (Rieger et al. (2022)) tackles backdoor attacks in FL by clustering the depth model of the last layer to filter outliers. CRFL (Xie et al. (2021)) employs clipping and smoothing to establish certified robustness against backdoor attacks. FLAME (Nguyen et al. (2022b)) adopts a weak Differential Privacy (DP) and dynamically clipped boundaries, effectively mitigating backdoor attacks while maintaining high accuracy on the main task. FEDCPA (Han et al. (2023)) evaluates the normality of local models and aggregates updates using weighted averaging, neutralizing the impact of potential malicious updates. GAS (Liu et al. (2023)) seeks to overcome the curse of dimensionality by partitioning high-dimensional gradients into low-dimensional subvectors, identifying trustworthy subvectors, and aggregating them to resolve the gradient heterogeneity problem. While these server-side defenses presume a lower number of malicious clients compared to benign clients during each global aggregation round, recent endeavors focus on enhancing the client's capability to defend against backdoor attacks during local training. FL-WBC (Sun et al. (2021)) and LeadFL (Zhu et al. (2023)) optimize the Hessian matrix to diminish the impact of backdoor features. FLIP (Eugene et al. (2020)) constructs and trains backdoor triggers against each other to counter backdoors. Lockdown (Huang et al. (2023b)) restricts client access to only a subset of model parameters, preventing toxic submodels from poisoning all parameters of the global model. Although effective in limiting backdoor impact, this method also diminishes the generalization performance of the global model.

## 3 BACKGROUND AND THREAT MODEL

**Federated learning.** In a traditional FL setup, $N$ clients collaborate to train a global model. Each client $i$ ($1 \le i \le N$) has its own data distribution $p(x_i, y_i)$ and a dataset $\mathcal{D}_i$ consisting of $N_i$ data points $\{(x_i^{(k)}), y_i^{(k)}\}_{k=1}^{N_i}$ covering $c_i$ classes, with $y_i^{(k)} \in c_i$ being its corresponding label. It is usually assumed that the data distribution $p(x_i, y_i)$ varies among the clients. Next, we denote the set of data from the N clients as $x = \{x_1, x_2, \cdots, x_N\}$ and the corresponding set of labels as $c = \{c_1, c_2, \cdots, c_N\}$. The global objective function to be minimized by AS is:

$$f(w) := \frac{1}{N} \sum_{i=1}^{N} f_i(w), \tag{1}$$

where $f_i(w) := \frac{1}{|N_i|} \sum_{(x_j^{(k)}, y_j^{(k)}) \in \mathcal{D}_i} \ell(w; x_j^{(k)}, y_j^{(k)}))$. According to the framework of representation learning, we divide the local model into a feature extractor (body) and a classifier (head), denoted

as $w_i = [w_i^b; w_i^{(h)}]$. Initially, we learn a low-dimensional representation $z$ of the original data $x_i$ with distribution $p(z|x_i; w_i^{(b)})$ parameterized by $w_i^{(b)}$. After extracting $z$, we train a classifier to generate a prediction of $z$ with a predictive distribution $\hat{p}(y_i|z; w_i^h)$ parameterized by $w_i^{(h)}$. Thus, for regression or classification tasks where the loss function is typically chosen to be a negative logarithmic prediction, the local objective of the client $i$ can be expressed as:

$$f_i(w_i) = \mathbb{E}_{p(x_i, y_i)} \left[ -\log \mathbb{E}_{p(z|x_i)}[\hat{p}(y_i|z)] \right] \tag{2}$$

To simplify the notation, we omit $w_i^{(h)}$ and $w_i^{(b)}$. Without loss of generality, we consider $p(x_i|z) = \mathcal{N}(z|\mu(x_i), \Sigma(x_i))$, where $\mathcal{N}$ denotes a normal distribution and $w_i^{(b)}$ produces both the mean $\mu$ and the covariance matrix $\Sigma$ of $z$. With the representation distribution explicitly modeled, we can easily constrain this known representation distribution.

**Threat model.** We assume that server will honestly follow the learning protocol. Clients are then classified into two categories: honest and malicious. Honest clients honestly execute the FL protocol, whereas malicious clients attempt to corrupt the global model by launching poisoning attacks locally. We consider the attacks outlined in previous studies (Chulin et al. (2020); Gu et al. (2017); Barni et al. (2019); Zhang et al. (2022)), assuming that the attacker can manipulate a group of compromised clients. To execute a backdoor attack, the attacker initially selects a backdoor trigger and a target class. Subsequently, in the $t$-th round ($t$=1,2,...,$T$) of communication, the compromised clients integrate triggers into a portion of local training data and reassign them as target classes. In SOTA backdoor attacks (Chulin et al. (2020); Gu et al. (2017); Barni et al. (2019); Zhang et al. (2022)), the attacker employs these training samples containing triggers to implant a backdoor into the compromised client's local model, thereby targeting the global model.

## 4 METHODOLOGY

Existing mainstream client selection methods and backdoor attack defenses select clients for aggregation based on model features. However, there is an inherent conflict between these two strategies. To address this issue, we select clients for aggregation based on data class representations rather than model features. Additionally, we incorporate data augmentation techniques to defend against attacks while enhancing local model performance, instead of using perturbation or clipping techniques that could degrade the local model's performance. Specifically, we first design a local model purification method on the client side that uses data augmentation techniques to alter the parameter distributions of local initial models, thereby protecting benign local models from backdoors. This method continuously changes the parameter distributions on the client side through data augmentation. When a few malicious local models are present during aggregation, their attack effects are weakened by benign models, making it difficult for them to impact the global model significantly. Next, we employ representation learning to capture class representations of local data for server-side client sampling. By identifying and selecting local models based on class representations rather than the models themselves, this approach avoids the curse of dimensionality and improves the proportion of benign models in the aggregated model. We then introduce a combination sampling method for class representations on the server side using upper confidence methods to minimize the loss of global class representations. Backdoor attacks tend to misclassify trigger-inserted samples as specific classes, causing their class representations to deviate significantly from the global class representation compared to benign models. Therefore, models are selected for aggregation based on the difference between local and global class representations. This method not only excludes malicious clients from participating in aggregation but also allows combining clean class representations to reduce the loss of global class features. The workflow is shown in Fig. 1.

### 4.1 LOCAL MODEL PURIFICATION

It is challenging for servers to aggregate only benign models with 100% accuracy. As a result, the global model and global representation can be contaminated to some extent. Especially in the later stages of training, even a single backdoored model can compromise the global model. This compromised global model is then sent to clients in subsequent rounds, causing benign local models

Figure 1: Overview of Grace. The client first downloads the global model from server. Once the client receives the global model, it begins by applying the local initial model purification method to eliminate the impact of the backdoor in the initial model. Following this, the client learns the local class representation and model using local representation learning algorithm. After completing the local training, the client uploads the local class representations and model updates to AS. Upon receiving these updates, server first selects the clients to participate in the aggregation using representation combination sampling. It then aggregates the selected clients' contributions to update the global model and global class representations.

to become gradually poisoned as well. Existing defenses attempt to identify backdoor-related parameters during local training, subsequently perturbing or clipping them. However, due to the vast local model parameter space, these defenses often mistakenly identify many benign parameters as malicious, leading to a significant degradation in local model performance. To address this issue, we propose a local initial model purification method to ensure that the local models of benign clients remain unaffected by backdoors while also improving their accuracy. Specifically, we first use data augmentation techniques to purify the initial local model and then apply local conditional mutual information (CMI) constraints to guide the learning of local class representations.

**Initial model purification.** Typically, an attacker creates $x'$ by making trivial modifications (i.e., adding a trigger $\delta$) to the clean data $x$. The backdoor is inserted by training the model to learn the mapping $x' \to y'$. Here, $(x', y')$ represents the toxic data. If we can change the mapping from $x' \to y'$ to $x' \to y$, we can obtain a reliably clean model instead of a backdoor-infected one. This is because, in this case, the model treats $\delta$ as an augmented feature and $x'$ as augmented clean data. In summary, by using $y$ instead of $y'$, we can turn the backdoor insertion process into a data augmentation process. Ideally, fine-tuning the backdoor model with toxic data paired with its corresponding ground truth labels (i.e., $(x', y)$) as augmented data can eliminate the backdoor. However, in reality, since we cannot get the actual backdoor triggers, we relax this process using data augmentation. Specifically, we create an augmented dataset using existing data augmentation techniques, such as Mixup (Zhang et al. (2018)). For Mixup, we can perform $\tilde{x}_{i,j} = \lambda x_i + (1 - \lambda)x_j$ and $\tilde{y}_{i,j} = \lambda y_i + (1 - \lambda)y_j$ for $\lambda \in [0, 1]$. Here, $\tilde{y}_{i,j}$ denotes a linear combination of the one-hot vectors corresponding to $y_i$ and $y_j$. The loss is defined as:

$$\ell^{\text{mix}}(\theta, \mathbb{D}_{\text{val}}) = \frac{1}{N_{\text{val}}^2} \sum_{i,j=1}^{N_{\text{val}}} \mathbb{E}_{\lambda \sim \mathcal{D}_\lambda} \ell(\tilde{y}_{i,j}, f_\theta(\tilde{x}_{i,j})) \tag{3}$$

where $\mathcal{D}_\lambda$ is a distribution on $[0, 1]$. In this work, we adopt the widely used $\mathcal{D}_\lambda$-Beta distribution, $Beta(\alpha, \beta)$, where $\alpha, \beta > 0$.

We observe that applying high-intensity local purification to the initial local model in each round does not effectively eliminate the impact of backdoor attacks and significantly degrades the performance of the main task. This situation can be likened to the human learning process: just as a child needs to learn to crawl before walking and running, skipping the crawling stage can impede knee development and adversely affect the child's ability to walk. Similarly, the server-client communication process reflects the stages of the global model's growth. In the early stages of communication, the global model is not yet stable. If data augmentation is employed too early, prompting the global model to learn mixed features, it may forget the data's primary features, resulting in a decline in main task accuracy that is difficult to recover. To mitigate this issue, we control the purification intensity by adjusting the number of local purification rounds. Specifically, we define $Np_i = r_i$, where $r_i$ represents the number of client-server communication rounds. When the client-server communication is in its early stages, the global model remains relatively weak, so we reduce the purification intensity, allowing the model to learn from augmented data over fewer rounds. As the communication

rounds increase, we gradually escalate the purification intensity, enabling the global model to adapt to learning more complex features. Our experimental analysis compares uniform purification with an increasing progressively purification intensity and shows that the global model achieves optimal performance when $Np_i = r_i$.

**Local representation learning.** To enable the server to recognize benign models, clients need to learn the class representations of their local data during local training. Inspired by Zhang et al. (2023b), we accomplish this process based on conditional mutual information (CMI) representation learning. Specifically, we enforce a constraint on local client updates, limiting the discrepancy between each client's local CMI $I_i(z; x_i|y_i)$ and the global CMI $I(z; x|y_i)$. The local (global) CMI quantifies the correlation between local (global) features and the input data $x_i$ given a specific label $y_i$. By integrating this information-theoretic constraint, the local class representation loss is defined as Zhang et al. (2023b):

$$
\begin{aligned}
\ell_i^{CMI} &= I(z; x|y_i) - I_i(z; x_i|y_i) \\
&= \mathbb{E}_{p(x,y_i)}\mathbb{E}_{p(x|x_i,y_i)}[KL[p(z|x) \| p(z|x_i)],
\end{aligned}
\tag{4}
$$

where $KL[p(z|x) \| p(z|x_i)]$ represents the Kullback-Leibler (KL) divergence, indicating the class-level feature alignment between $p(z|x)$ and $p(z|x_i)$ for a given label $y_i$. Eq. 4 effectively limits the KL divergence between $p(z|x)$ and $p(z|x_i)$, ensuring that the stochastic representations of the joint and individual posteriors within the class remain consistent across clients.

In optimization theory, the L2-norm of a model parameter is frequently used to measure model complexity. Similarly, we introduce an L2R on the class representation, rather than on the network parameters. This serves to constrain the complexity of the class representation and further enhance the stability of the local class representation. The regularizer is expressed as Nguyen et al. (2022a):

$$
\ell_i^{L2R} = \mathbb{E}_{p_i(x)}\left[\mathbb{E}_{p(z|x)}[||z||_2^2]\right] \approx \frac{1}{N_i}\sum_{n=1}^{N_i} ||z_i^{(n)}||_2^2,
\tag{5}
$$

where $z_i^{(n)}$ is a single sample from $p(z|x_i^{(n)})$.

Thus, the local optimization objective for client $i$ is

$$
\ell_i = f_i + \alpha^{L2R}\ell_i^{L2R} + \alpha^{CMI}\ell_i^{CMI}.
\tag{6}
$$

## 4.2 REPRESENTATION COMBINATION SAMPLING

**Global class representation loss.** Accurately identifying malicious clients has always been a challenging problem in defending against backdoor attacks in FL. Existing defenses typically detect distinctiveness introduced by backdoor data from a model perspective. However, due to the curse of dimensionality and model complexity, these defenses often struggle to identify backdoor models effectively. To address this issue, we design a metric called Global Class Representation Loss (*GCRL*) to select benign models for aggregation based on class representations, rather than relying on the model perspective. The specific definition is as follows:

$$
GCRL(\mathcal{M}) \triangleq \sum_{c=1}^{C} KL[p_n(z)||p(z)].
\tag{7}
$$

It is worth noting that we design *GCRL* with the same loss function as the local class representation loss. This design allows Grace to defend against backdoor attacks while also selecting local models that enhance FL performance for aggregation. First, let's explain why *GCRL* is effective in defending against backdoors. GCRL measures the difference between local and global class representations. For benign models, their local class representations are more similar to the global class representations, resulting in a smaller difference. However, malicious clients have the additional task of embedding backdoors, causing their class representations to consistently differ from those of benign models during training. This leads to a larger difference between the class representations of malicious clients and the global class representation. Therefore, using *GCRL* in the aggregation process effectively excludes malicious models. Next, let's discuss why *GCRL* can improve FL performance. For a benign client, a smaller difference between its local and global class representations suggests that the client's local data is more balanced and that it has learned most of the class features. Existing studies indicate that imbalances in local data classes degrade FL performance. Therefore, using *GCRL* as a criterion for selecting benign clients helps improve the overall performance of FL.

**Combination sampling.** To identify the most representative dataset $\mathcal{D}_{\mathcal{M}}$, we aim to determine the optimal subset $\mathcal{M}^*$ by minimizing the *GCRL*, defined as follows:

$$\mathcal{M}^* \triangleq \arg\min_{\mathcal{M} \subseteq \{1,2,\cdots,N\}} GCRL(\mathcal{M}). \tag{8}$$

The primary challenge lies in computational complexity. To search the exact optimal $\mathcal{M}^*$, it's necessary to iterate through all possible scenarios to identify the lowest *GCRL* value. The computational complexity scales as $\mathcal{O}\left(\binom{N}{M} \times M^2\right)$, which becomes impractical as $N$ grows large.

To handle the computational challenge, rather than treating $\mathcal{M}$ as a fixed set, we consider it as a sequence of random variables $M = \{S_1, S_2, \cdots, S_i, \cdots, S_M\}$, each with a probability assignment. Essentially, if $\mathcal{M}$ tends to have lower class representation loss, it should be more likely to be sampled.

Our strategy involves sequential element generation in $\mathcal{M}$. Initially, we sample $c_1$ based on the probability $P(S_1 = s_1)$. Subsequently, we sample $s_2$ considering the conditional probability $P(S_2 = s_2|S_1 = s_1)$ to form $\mathcal{M}_2 = \{s_1, s_2\}$. This process continues, iteratively selecting clients, until we construct $M = \{S_1, S_2, \cdots, S_M\}$. In the following, we'll define appropriate conditional probabilities to align with our expectations in choosing clients.

We use $T_k$ to denote the number of times a client $k$ is selected. When the client $k$ is selected in a communication round, $T_k$ increments by 1; otherwise, $T_k$ remains unchanged. Our approach draws inspiration from the combined upper confidence boundary (CUCB) algorithm (Chen et al. (2013)) and prior researches (Zhang et al. (2023c); Yang et al. (2021a)). The probability of selecting the first client in the $t$-th round of communication is determined by:

$$P(S_1 = s_1) \propto \frac{1}{[GCRL(\mathcal{M}_1)]^{\gamma_1}} + \lambda \sqrt{\frac{3\ln t}{2T_{s_1}}}, \gamma_1 > 0. \tag{9}$$

In the above equation, $\lambda$ plays a crucial role in balancing exploitation and exploration. The second term amplifies the likelihood of selecting clients that haven't been sampled in previous communications. Following the selection of $S_1$, the probability of sampling the second client is formulated as:

$$P(S_2 = s_2|S_1 = s_1) \propto \frac{\frac{1}{[GCRL(\mathcal{M}_1)]^{\gamma_2}}}{\frac{1}{[GCRL(\mathcal{M}_1)]^{\gamma_1}} + \lambda \sqrt{\frac{3\ln t}{2T_{s_1}}}}, \gamma_2 > 0. \tag{10}$$

The probability of selecting the $i$-th client, where $2 < i \le M$, is defined as follows:

$$P(S_i = s_i|S_{i-1} = s_{i-1}, \cdots, S_2 = s_2, S_1 = s_1)$$
$$\propto \frac{[GCRL(\mathcal{M}_{i-1})]^{\gamma_{i-1}}}{[GCRL(\mathcal{M}_i)]^{\gamma_i}}, \gamma_{i-1}, \gamma_i > 0. \tag{11}$$

In this sampling process, the ultimate probability of selecting $\mathcal{M}$ is denoted as $P(S_1 = s_1, S_2 = s_2, \cdots, S_M = s_M) = P(S_1 = s_1) \times P(S_2 = s_2|S_1 = s_1) \cdots \times P(S_M = s_M|S_{M-1} = s_{M-1}), \cdots, S_2 = s_2, S_1 = s_1 \propto \frac{1}{[GCRL(\mathcal{M})]^{\gamma_M}}$. Given that $\gamma_M > 0$, this aligns with our objective that $\mathcal{M}$ with a lower value *GCRL* should have a higher sampling probability. The client has to align its local class representation with clean global class representation. It raises a critical question: How can AS obtain a comprehensive global class representation $p(z|x)$ of each class without direct access to the client's raw data? Since clients are reluctant to disclose the raw data, we introduce the product of experts (PoE) (Hinton (2002)), which decomposes the joint posterior into a product of individual posteriors.

$$p(z|x) = p(z|x_1, \cdots, x_m) \propto \tau p(z) \prod_{i=1}^{m} p(z|x_i), \tag{12}$$

where $\tau = \frac{\prod_{i=1}^{M} p(x_i)}{p(x_1,\cdots,x_m)}$ indicates the degree of independence among clients, and $p(z)$ represents a prior distribution, often modeled as a spherical Gaussian.

Using PoE, a straightforward analytical solution arises when $p(z|x_i)$ takes the form of a diagonal Gaussian distribution. Thus, we can obtain the product of Gaussian experts, which consists of mean $\mu = (\mu_0 \Sigma_0^{-1} + \sum_{i \in M} \mu_i \Sigma_i^{-1})(\Sigma_0^{-1} + \sum_{i \in M} \Sigma_i^{-1})^{-1}$ and covariance $\Sigma = (\Sigma_0^{-1} + \sum_{i \in M} \Sigma_i^{-1})^{-1}$, where $p(z|x_i) \sim \mathcal{N}(\mu_i, \Sigma_i)$ and $p(z) \sim \mathcal{N}(\mu_0, \Sigma_0)$.

## 5 EXPERIMENT

**Attack method.** We assess our defense against three data-level backdoor attacks and an advanced backdoor attack, following Lockdown's (Huang et al. (2023b)) attack setup. The data-level attacks include BadNets (Gu et al. (2017)), DBA (Chulin et al. (2020)), and Sinusoidal (Barni et al. (2019)). Additionally, the advanced backdoor attack is Neurotoxin (Zhang et al. (2022)). Further details can be found in Appendix A.3.

**Defense baseline.** We compare Grace with nine baselines, namely FedAvg (McMahan et al. (2017)) without any defense, FLAME (Nguyen et al. (2022b)), FLTrust (Cao et al. (2021)), RFA (Pillutla et al. (2022)), FedCPA (Han et al. (2023)), LeadFL (Zhu et al. (2023)) and Lockdown (Huang et al. (2023b)). FedAvg serves as an undefended baseline to evaluate the effect of defense on global model accuracy. All approaches are implemented based on open-source code.

**Selection baseline.** We utilize randomly selected FedAvg as our baseline and compare Grace with three SOTA selection schemes: DivFL (Balakrishnan et al. (2022)), FedCBS (Zhang et al. (2023c)), and Fed-cucb (Yang et al. (2021b)).

**Evaluation metrics.** We primarily use the primary task accuracy (ACC) and the attack success rate (ASR) of the global model as evaluation metrics. To assess the effectiveness of excluded, we measure the proportion of malicious models (PMM) participating in aggregation and the accuracy of poison samples (PACC) as clean labels. All experiments are conducted using two Nvidia 4090 GPUs. Due to the space limit, we moved a detailed description of the experiment setup and additional results to the Appendix.

### 5.1 MAIN EVALUATION

**Defense effectiveness.** In our main evaluation, we utilize CIFAR-10 as the default dataset and BadNets as the default attack for assessing the performance of defenses against backdoor attacks.

The defense effectiveness results for communication rounds are depicted in Fig. 2. Grace exhibits the strongest robustness in both IID and non-IID attack settings. Specifically, the ASR of Grace (indicated by the red line) consistently approaches 0% in both settings, while its ACC remains consistently above 80% after 100 rounds. In Figs. 2(b) and (d), Grace exhibits a 100% ASR in the initial rounds due to the RCS method requiring each client to participate in aggregation once at the start of training. Consequently, there are certain rounds with a higher number of attackers, leading to a 100% ASR. However, after all clients have participated in aggregation, the ASR tends to stabilize, particularly in the IID setting, where it remains close to 0%. Compared to FedAvg, Grace reduces the ASR by 98% in IID settings and by 93% in non-IID settings. Besides, the learning curves of existing defenses for both ACC and ASR show significant fluctuations, indicating instability. Specifically, in Figs. 2(b) and (d), the ASR learning curves of existing defenses vary widely, ranging from 0 to 100, with more pronounced instability in non-IID settings. In the IID setting, only FedCPA's ASR approaches that of Grace; however, FedCPA's ACC shows substantial fluctuations in the non-IID setting, indicating a lack of stability.

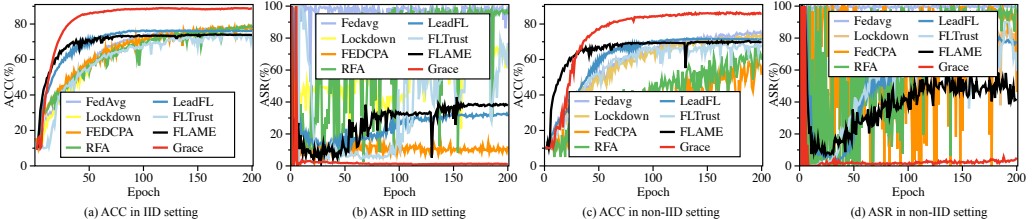

Figure 2: ACC and ASR under different defense.

**Selection effectiveness.** For evaluating the performance of the client selection scheme, which includes the comparison term of feature distribution skew, we utilize FEMNIST, while all other evaluations are conducted using CIFAR-10. To evaluate the effectiveness of the selection, we set the number of malicious clients to 0. We assess Grace under IID and five heterogeneous settings: quantity-based labeling imbalance (Label#C2), distribution-based labeling imbalance (non-IID), noise-based feature imbalance (Feature-N), real-world feature imbalance (Feature-R), and quantity imbalance (Quantity). Feature-R is constructed using the FEMNIST dataset, while CIFAR-10 is used for all other cases. As shown in Table 1, the non-IID and Label#C2 settings are more complex than the

other partitioning methods, making federated learning more challenging. In particular, under the Label#C2 setting, the ACC of FedAvg is only 54.83%. In comparison to existing solutions, Grace exhibits superior ACC in more complex settings. Specifically, Grace improves ACC by 11.24% and 17.31% over FedAvg for the non-IID and Label#C2 settings, respectively. Additionally, Grace achieves optimal ACC across four heterogeneous settings, while FedCBS attains optimal ACC in only two settings.

To demonstrate that Grace's selection strategy can effectively exclude malicious models during aggregation, we present PMM included in each round of aggregation under two complex heterogeneous settings: non-IID and Label#C2. Among existing backdoor defense schemes, only FLAME selectively aggregates benign clients. Additionally, to highlight the limitations of using existing client selection methods for defending against backdoor attacks, we show the PMM included in each round of aggregation for these methods. As shown in Fig. 3, the PMM in the FLAME scheme gradually rises to 0.6, indicating that identifying malicious clients based solely on the similarity of local model parameters is not reliable. For existing client selection schemes such as Fed-cucb, Fed-CBS, and DivFL, the proportion of malicious clients fluctuates around 0.4, which corresponds to the preset ratio of malicious clients to the total number of clients. In contrast, Grace maintains a proportion of malicious clients below 0.2 in most rounds, demonstrating its ability to efficiently exclude malicious models during aggregation.

Table 1: ACC(%) under different data partitioning methods

| Partitioning | FedAvg | Fed-cucb | DivFL | Fed-CBS | Grace |
|---|---|---|---|---|---|
| IID | 86.17 | 86.82 | 87.54 | 88.41 | **88.66** |
| non-IID | 74.33 | 75.46 | 82.63 | 83.41 | **86.00** |
| Lable#C2 | 54.83 | 55.73 | 67.84 | 69.32 | **72.14** |
| Feature-N | 83.16 | 82.51 | 84.98 | **85.63** | 85.46 |
| Feature-R | 87.51 | 88.35 | 89.16 | **90.25** | 89.86 |
| Quantity | 87.37 | 87.45 | 89.98 | 90.05 | **90.45** |

Figure 3: Excluded effectiveness.

**Defense effectiveness under varying attacker ratio.** As shown in the Fig. 4, Grace's ASR is significantly lower than existing defenses for different attacker ratios. Meanwhile, Grace's ACC is significantly higher than FedAvg, which is not achieved by existing defenses. Notably, the variations in Grace's ACC and ASR are minimal even as the attacker ratio increases, demonstrating Grace's robustness. We then observe the existing defenses, whose ACC fluctuates significantly as the number of attackers increases in the non-IID setting. Besides, the ASR of the existing defenses, both in the IID and non-IID settings, shows jumps as the number of attackers changes. This further highlights the advancement of Grace.

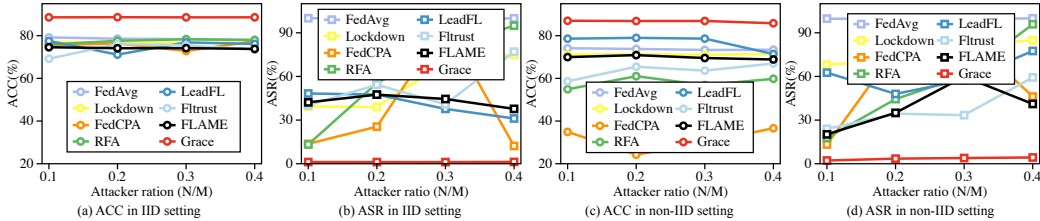

Figure 4: ACC and ASR under different attacker ratio.

**Defense effectiveness under varying poison ratio** As depicted in Table 2, Grace's ASR is significantly lower than the existing defenses, and its ACC is significantly higher than the existing defenses for varying poisoning rates. Furthermore, Grace's ACC and ASR remain stable as the poisoning rate increases, demonstrating its robustness to different levels of poisoning. As the poisoning rate increases, the ACC of FedAvg exhibits a declining trend. Its decline is attributed to the reduction in benign data, which diminishes the accuracy of the local model, consequently leading to a decrease in the ACC of the global model. Grace's ACC also shows this pattern, but the decline in Grace is very slight, less than 1%. In contrast, the ACC of existing defenses does not follow this pattern because their defensive processes tend to sacrifice ACC. Existing defenses do not exhibit a clear pattern of ASR change as the poisoning rate increases, indicating their difficulty in accurately identifying malicious parameter spaces or models, leading to inconsistent performance. Moreover, in

IID settings, only FLAME and LeadFL consistently reduce ASR below 50%. However, in non-IID settings, none of the existing defenses consistently achieve an ASR below 50%.

Table 2: Defense effectiveness under different poison ratio $p$

| Methods (IID) | ACC(%) ↑ | | | | ASR(%) ↓ | | | |
|---|---|---|---|---|---|---|---|---|
| | $p$=0.1 | $p$=0.3 | $p$=0.5 | $p$=0.8 | $p$=0.1 | $p$=0.3 | $p$=0.5 | $p$=0.8 |
| FedAvg (McMahan et al. (2017)) | 78.82 | 78.43 | 78.18 | 75.66 | 100 | 99.99 | 99.99 | 97.48 |
| Lockdown (Huang et al. (2023b)) | 76.94 | 76.16 | 76.50 | 75.62 | 48.92 | 58.57 | 74.91 | 88.61 |
| FedCPA (Han et al. (2023)) | 78.23 | 76.52 | 77.39 | 75.88 | 93.79 | 91.03 | 12.21 | 97.05 |
| RFA (Pillutla et al. (2022)) | 78.65 | 78.16 | 77.91 | 75.38 | 78.00 | 82.97 | 94.88 | 85.41 |
| LeadFL (Zhu et al. (2023)) | 75.22 | 76.07 | 76.14 | 74.01 | 42.7 | 36.1 | 31.13 | 32.8 |
| FLTrust (Cao et al. (2021)) | 74.76 | 68.85 | 73.46 | 73.14 | 52.9 | 22.7 | 77.01 | 38.9 |
| FLAME (Nguyen et al. (2022b)) | 73.38 | 73.43 | 73.81 | 74.72 | 25.32 | 29.71 | 37.72 | 44.46 |
| Grace | **88.77** | **88.58** | **88.62** | **88.36** | **1.33** | **1.21** | **1.24** | **1.47** |
| Methods (non-IID) | ACC(%) ↑ | | | | ASR(%) ↓ | | | |
| | $p$=0.1 | $p$=0.3 | $p$=0.5 | $p$=0.8 | $p$=0.1 | $p$=0.3 | $p$=0.5 | $p$=0.8 |
| FedAvg (McMahan et al. (2017)) | 74.22 | 73.21 | 73.41 | 71.43 | 100 | 100 | 82.52 | 97.99 |
| Lockdown (Huang et al. (2023b)) | 72.66 | 73.27 | 72.91 | 70.95 | 52.14 | 67.72 | 81.79 | 89.26 |
| FedCPA (Han et al. (2023)) | 68.21 | 66.02 | 51.38 | 33.65 | 85.93 | 33.12 | 46.18 | 78.13 |
| RFA (Pillutla et al. (2022)) | 62.73 | 59.44 | 59.83 | 56.19 | 59.62 | 54.38 | 95.85 | 99.6 |
| LeadFL (Zhu et al. (2023)) | 71.96 | 73.06 | 71.41 | 70.13 | 53.11 | 61.83 | 77.50 | 71.67 |
| FLTrust (Cao et al. (2021)) | 65.33 | 68.91 | 66.97 | 66.63 | 31.00 | 43.30 | 59.31 | 51.90 |
| FLAME (Nguyen et al. (2022b)) | 68.26 | 67.09 | 68.85 | 64.20 | 46.7 | 58.20 | 41.17 | 52.98 |
| Grace | **85.93** | **85.86** | **85.83** | **85.64** | **4.41** | **4.21** | **4.95** | **6.53** |

**Defense effectiveness under different data partitioning methods.** In addition to utilizing the Dirichlet distribution for client data partitioning, we delve into other strategies to partition client data in a manner that is more closely with practical scenarios. Detailed descriptions of these methods are provided in the Appendix A.4. The results are shown in Table 3. They demonstrate that Grace achieves SOTA defense under various data partitioning strategies, consistently maintaining defense performance below 5%. We observe that the ACC of FedAvg drops to 52.34% in the Label#C=2 setting, while the ACCs of existing defenses are all below 45%. Compared to FedAvg, Grace shows an improvement of 19.55% in ACC. Although existing defenses like Foolsgold, RFA, and FLTrust reduce ASR to 0% under the Label#C=2 setting, their ACCs are below 30%.

Table 3: Defense effectiveness under different data partitioning methods

| Methods | ACC(%) ↑ | | | | ASR(%) ↓ | | | |
|---|---|---|---|---|---|---|---|---|
| | IID | non-IID | Label#C=2 | Quantity | IID | non-IID | Label#C=2 | Quantity |
| FedAvg (McMahan et al. (2017)) | 78.18 | 73.41 | 52.34 | 86.91 | 99.99 | 82.52 | 99.81 | 100.0 |
| Lockdown (Huang et al. (2023b)) | 76.50 | 72.91 | 51.91 | 85.79 | 74.91 | 81.79 | 61.31 | 83.20 |
| FedCPA (Han et al. (2023)) | 77.39 | 51.38 | 19.34 | 85.19 | 12.21 | 46.18 | 50.44 | 97.15 |
| RFA (Pillutla et al. (2022)) | 77.91 | 59.83 | 17.41 | 86.62 | 94.88 | 95.85 | 0 | 97.67 |
| LeadFL (Zhu et al. (2023)) | 76.14 | 71.41 | 51.41 | 84.50 | 31.13 | 77.50 | 69.04 | 45.22 |
| FLTrust (Cao et al. (2021)) | 73.46 | 66.97 | 26.04 | 62.12 | 77.01 | 59.31 | 0 | 23.92 |
| FLAME (Nguyen et al. (2022b)) | 73.81 | 68.85 | 37.38 | 59.33 | 37.72 | 41.17 | 19.8 | 16.80 |
| Grace | **88.62** | **85.83** | **71.89** | **89.64** | **1.24** | **4.95** | 4.37 | **2.94** |

# 6 CONCLUSION

In this study, we propose a client selection method that supports backdoor attack defense, addressing the conflict between existing client selection strategies and backdoor defenses. Specifically, we propose a local model purification method on the client side to enhance the robustness of the local models against attacks. On the server side, we develop representation combination sampling methods to select clients for aggregation based on local data class representations, effectively excluding malicious clients while also identifying high-quality ones. Our experimental results demonstrate that Grace not only achieves SOTA defense but also improves the performance of the global model, marking a significant advancement over existing defenses that tend to degrade model performance. Looking ahead, we acknowledge the potential for attackers to optimize their strategies based on data representations, underscoring the need for further investigation in our future research endeavors.

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
