# OpenReview forum: "A robust federated learning client selection with combinatorial data class representations and data augmentation"
_ICLR.cc/2025/Conference — ICLR 2025 Conference Withdrawn Submission_

### Official Review · Reviewer_UiVx · 2024-11-01

**Soundness:** 2
**Presentation:** 3
**Contribution:** 2
**Rating:** 5
**Confidence:** 3

**Summary:**

This study proposes a client selection method to support defenses against backdoor attacks. Specifically, it enhances robustness against attacks by using a local model purification method on the client side. On the server side, it develops a method to select clients based on the class representation of local data, identifying high-quality clients while excluding malicious ones.

**Strengths:**

This study is the first solution that defends against backdoor attacks without sacrificing model accuracy. It surpasses existing defenses under different attack scenarios and non-IID settings.

**Weaknesses:**

The threat model setup and the limitations of this study are not clearly defined, and comprehensive experimental evaluation is lacking. The threat model does not mention Byzantine attacks or the bypassing of local processing. This study only addresses horizontal federated learning, with no mention of vertical federated learning. In the experimental evaluation, mainly CIFAR-10 is used, and the impact of different numbers of clients is unclear.

**Questions:**

How does the threat model consider Byzantine attacks? Is there a concern that local processing might be bypassed or that compromised models might be uploaded? Does this study only address horizontal federated learning? Are there any limitations to applying it to vertical federated learning?

---

### Official Review · Reviewer_21jZ · 2024-11-03

**Soundness:** 1
**Presentation:** 1
**Contribution:** 1
**Rating:** 3
**Confidence:** 5

**Summary:**

This paper proposes a client selection framework, "Grace," designed to enhance federated learning (FL) security by addressing vulnerabilities to backdoor attacks while improving model accuracy. Grace introduces two main components: local model purification and a representation combination sampling method. The local model purification technique uses data augmentation to improve model robustness on the client side, reducing the risk of backdoor contamination. On the server side, Grace employs combinatorial class representations to select clients for aggregation based on data characteristics, helping to filter out potentially malicious clients. Experimental results show that Grace significantly reduces the attack success rate compared to state-of-the-art defenses while maintaining higher model accuracy across various non-IID data distributions.

**Strengths:**

This paper introduces Grace, which aims to solve multiple challenges simultaneously via the client selection.

**Weaknesses:**

1. The target of this paper seems not clear. In the title, the paper emphasizes client selection; however, in the abstract, the authors focus on enhanced defense performance. Still, in the contribution, the authors also mentioned the attacks in non-IID problems, whereas the whole paper did not mention what are the exact non-IID settings.
2. The baseline selection is out-of-date. Recently, new backdoor attacks were published in 2023 and 2024, and this paper needs to have an evaluation of the SOTA attack and defences methods to evaluate the effectiveness of the proposed method. Such as:
[1] Zhang, H., Jia, J., Chen, J., Lin, L. and Wu, D., 2024. A3fl: Adversarially adaptive backdoor attacks to federated learning. Advances in Neural Information Processing Systems, 36.
[2] Nguyen, T.D., Nguyen, T.A., Tran, A., Doan, K.D. and Wong, K.S., 2024. Iba: Towards irreversible backdoor attacks in federated learning. Advances in Neural Information Processing Systems, 36.
[3] Huang, T., Hu, S., Chow, K.H., Ilhan, F., Tekin, S. and Liu, L., 2024. Lockdown: backdoor defense for federated learning with isolated subspace training. Advances in Neural Information Processing Systems, 36.
[4] Kumari, K., Rieger, P., Fereidooni, H., Jadliwala, M. and Sadeghi, A.R., 2023, May. Baybfed: Bayesian backdoor defense for federated learning. In 2023 IEEE Symposium on Security and Privacy (SP) (pp. 737-754). IEEE.
3. The proposed methods are confusing. The authors did not discuss how the data augmentation works, especially as stated in the title.
4. The selection methods based on local representation learning with global class representation loss seem to directly kick the benign clients out of the FL with minority classes.

**Questions:**

1. What are the non-IID settings of this paper, as the authors stated in the contribution?
2. More experiments are needed to compare with the SOTA methods in 2023 - 2024.
3. Authors claimed that Grace is robust to non-IID backdoor attacks, however, the method seems to kickout the non-IID clients directly in the FL to improve the overall performance.

---

### Official Review · Reviewer_BuyA · 2024-11-04

**Soundness:** 2
**Presentation:** 2
**Contribution:** 2
**Rating:** 3
**Confidence:** 4

**Summary:**

This paper presents "Grace”, a novel framework for federated learning that defends against backdoor attacks by effectively selecting client parameters for aggregation, based on their validation utilizing their class representations. In particular, Grace introduces a two-stage solution: purification of the local model is achieved by training with augmented data to weaken potential backdoor attacks, and a representation combination sampling technique to select honest clients based on class representations. Experimental results demonstrate that Grace not only achieves state-of-the-art defense but also improves the performance of the global model.

**Strengths:**

1. This paper proposes an innovative approach for federated learning that improves accuracy while defending against backdoor attacks.
2. The framework maintains client privacy while enabling server to effectively select honest and good quality clients for aggregation.
3. The results demonstrate that the proposed method maintains a stable performance with increasing poisoning rates.

**Weaknesses:**

1. Lack of convergence proof and theoretical supports. It would be helpful if the author formally define upper bound of the CMI component and prove convergence guarantee for GCLR.
2. Lack of experimental support for different heterogeneous settings and varying FL scenarios. For example, the authors may consider different levels of data heterogeneity (i.e., α=0.05,0.5,0.1)
3. The presentation quality of the paper needs to be improved. In particular, the authors may provide an algorithm formally in section 4, highlighting the key steps involved in the proposed approach.

**Questions:**

1. How robust is the system when majority of the clients are malicious?
2. Is this approach effective in case of extreme non-IID settings?
3. The authors have mentioned that the data augmentation technique is used to purify local model by remapping x'->y' to x'-y. What if y is not entirely present in the client dataset. It would be interesting if the authors can conduct experiments on some extreme non-IID setting and validate its impact.
4. Also, we suggest the authors to include more diverse datasets and scenarios in their experiment.

---

### Official Review · Reviewer_PGH1 · 2024-11-10

**Soundness:** 3
**Presentation:** 3
**Contribution:** 1
**Rating:** 5
**Confidence:** 5

**Summary:**

This paper presents Grace, a resilient client selection framework for federated learning (FL) that addresses the challenges posed by client heterogeneity and backdoor attacks. Traditional client selection and backdoor defenses often conflict, but Grace combines combinational class sampling and data augmentation to overcome this. On the client side, it introduces a local model purification method to enhance model robustness and extract local class representations, allowing secure client selection without compromising privacy. On the server side, Grace applies a novel sampling method based on these class representations to exclude malicious clients and select those with valuable data. Experiments demonstrate that Grace can achieve notable improvements in both security and accuracy.

**Strengths:**

1. The idea is easy to understand.

2. Many attacks/defenses are tested during the evaluation.

**Weaknesses:**

1. My main concern with the paper is the novelty issue. Since robust federated learning is already well-studied, the paper's contribution feels incremental. It appears to simply combine several technical concepts, such as representation learning and data augmentation, into its proposed methodology. However, this approach does not significantly advance the qualitative understanding of the security and privacy of FL in the machine learning domain.

2. The paper is neither well-written nor well-motivated, making it difficult to understand what led the authors to combine these concepts for robust federated learning client selection.

3. Additionally, the threat model is unclear, leaving the assumptions about attackers ambiguous. While Section 2 (Threat Model) references numerous existing works, it fails to discuss their relationship to the proposed threat model.

4. The evaluation results do not seem to support the authors' conclusion that "the results are compelling: ... " For example, as shown in Table 1, the proposed method underperforms compared to existing work, such as Fed-CBS, when evaluated with feature-based data partitioning methods. This discrepancy is not adequately explained in the evaluation section

5. Writing needs to be improved. Please proofread the whole paper carefully to correct typos and grammar errors.

**Questions:**

Please refer to my comments for more details.

---

### Note · Authors · 2024-11-21

I have read and agree with the venue's withdrawal policy on behalf of myself and my co-authors.